# Large-Scale Production of Size-Adjusted β-Cell Spheroids in a Fully Controlled Stirred-Tank Reactor

**Florian Petry and Denise Salzig \***

Institute of Bioprocess Engineering and Pharmaceutical Technology, University of Applied Sciences Mittelhessen, Wiesenstrasse 14, 35390 Giessen, Germany; florian.petry@lse.thm.de

\* Correspondence: denise.salzig@lse.thm.de; Tel.: +49-641-309-2630; Fax: +49-641-309-2553

**Abstract:** For β-cell replacement therapies, one challenge is the manufacturing of enough β-cells (Edmonton protocol for islet transplantation requires 0.5–1 $\times$ $10^6$ islet equivalents). To maintain their functionality, β-cells should be manufactured as 3D constructs, known as spheroids. In this study, we investigated whether β-cell spheroid manufacturing can be addressed by a stirred-tank bioreactor (STR) process. STRs are fully controlled bioreactor systems, which allow the establishment of robust, larger-scale manufacturing processes. Using the INS-1 β-cell line as a model for process development, we investigated the dynamic agglomeration of β-cells to determine minimal seeding densities, spheroid strength, and the influence of turbulent shear stress. We established a correlation to exploit shear forces within the turbulent flow regime, in order to generate spheroids of a defined size, and to predict the spheroid size in an STR by using the determined spheroid strength. Finally, we transferred the dynamic agglomeration process from shaking flasks to a fully controlled and monitored STR, and tested the influence of three different stirrer types on spheroid formation. We achieved the shear stress-guided production of up to $22 \times 10^6 \pm 2 \times 10^6$ viable and functional β-cell spheroids per liter of culture medium, which is sufficient for β-cell therapy applications.

**Keywords:** spheroid strength; β-cells; diabetes; shear stress-guided production; hydrodynamic stress





## 1. Introduction

Diabetes involves the selective autoimmune destruction or dysfunction of insulin-producing β-cells, located within the islets of Langerhans in the pancreas. The large and growing number of patients living with diabetes [1] has generated interest in the promise of β-cell therapy to restore lost β-cell mass [2]. However, β-cells have unique characteristics and must be transplanted as spheroids to be able to exert their full biological activity in the recipient. The β-cell spheroids are living drugs, and proper manufacturing is an important step to bring these therapeutics into clinics. It is difficult to manufacture sufficient β-cell numbers. The total estimated number of β-cells in the human pancreas is ~$10^9$ [3], which is the benchmark for the manufacturing process. Islets of Langerhans from deceased donors show a high functionality, but there is a shortage of donor material, the adult β-cells lose their functionality over time, and they cannot be expanded in vitro [4,5]. Induced pluripotent stem cells (iPSCs) can help to address the material shortage, but iPSCs lack functionality compared with native islets and require a complex and laborious differentiation protocol [4–6]. Various mouse and rat models have been used for diabetes research, which led to the development of rodent β-cell lines such as RIN (rat), INS-1 (rat) and MIN6 (mouse) [4]. However, the development of human β-cell lines has been hampered by their insufficient functionality and xenotropic viral contamination [5,7–9]. Even so, β-cell lines can be expanded in vitro over multiple passages without decreasing in functionality, therefore, β-cell lines serve as good models for diabetes research, and the development of manufacturing protocols for cell therapy [10,11]. The 3D cultivation of β-cell lines as

spheroids or agglomerates (these terms are used synonymously) can enhance glucose-dependent insulin secretion, and β-cells cultured as re-aggregated 3D structures (pseudo-islets) show a greater viability, proliferation and functionality than individual cells [12,13]. Such agglomerates facilitate the reconstitution of the native pancreatic microenvironment, including cell–cell interactions, the extracellular matrix (ECM), cell coupling and tight junctions, cell polarization, and changes in gene expression triggered by bioactive molecules and forces acting on the cytoskeleton [14–16].

A manufacturing process for cell therapy must produce uniform spheroids in sufficient numbers. As mentioned above, the benchmark for β-cell therapy is to manufacture ~$10^9$ cells [3] or $1–3.2 \times 10^6$ spheroids, following the cell and islet amount within the human pancreas [15,17]. Although the mean islet size is 140 μm [3], a major portion of islets ranges between 20 and 50 μm (37%) [18], or 10 and 50 μm (60%) [19]. Even though the contribution of the smaller islets to the total islet mass is small, they might be crucial for the outcome of the transplants, as smaller islets are more robust against hypoxic conditions [20]. Ultimately, this results in an increased functionality, as shown for islets < 100 μm in comparison with islets > 250 μm [21–23], therefore, we aim to manufacture spheroids between 20 and 100 μm. Small-scale systems, such as microtiter plates, can generate enough spheroids for research purposes, but they are labor-intensive and vulnerable to contamination, they lack process control, and they are difficult to scale up to produce sufficient cell quantities for clinical applications. Some groups address the scale-up issue by using spinner flasks [24–27] or rotating wall vessel reactors (RWVRs) [22,28], which enable a limited degree of process control and increase the manufacturing scale up to 500 mL. However, these systems achieve improper mixing and have limited possibilities to adjust the spheroid size. Our literature search did not reveal any reports regarding the large-scale (1 L or more) or stirred-tank bioreactor (STR) related expansion of β-cell spheroids, but STR results were found for other cell types (mainly iPSCs). Table 1 summarizes the critical aspects of the dynamic production of spheroids with different cell types in relation to our presented theoretical background (see Supplementary). STRs are fully controlled bioreactor systems, which allow the establishment of robust, larger-scale manufacturing processes. STRs also facilitate the online monitoring and control of process parameters such as $O_2$, pH, temperature, and biomass (the latter by dielectric spectroscopy). STR processes can also be automated and are scalable up to 6000 L (single-use and multi-use systems). Particularly, the scale-up of β-cell manufacturing is a crucial step. When transferring bioprocesses to larger scales, mass transport problems often occur and become process-limiting, which could not be observed or were negligible on a small scale. To avoid process limitations, a correct scale-up strategy is needed, for which the similarity theory provides good service. This theory uses dimensionless numbers (such as the stirrer power number, $N_P$, or the Reynolds number, Re) for the description of a physical–technical behavior of the manufacturing system (e.g., the STR), which is to be maintained constant during the transition to another scale, so that the physical similarity remains [29]. In the context of β-cell spheroid manufacturing, we see STRs as a potent choice of scalable bioreactor systems, as STRs provide geometric similarity ($d_s/D_T$, $H_L/D_T$, $d_S/d_{H,}$) at different scales, and the development of a turbulent flow, which enables a constant $N_P$ and the admissible application of Kolmogorov's theory of isotropic turbulence (see theoretical background or [30]).

Although we consider the STR as a proper system for β-cell spheroid manufacturing, we must be aware of the fact that the formation of cell agglomerates differs from the formation in static culture systems. Agglomeration in static cultures involves the self-assembly of cells in hanging drops or parabolic wells with cell-repellent surfaces [31]. In contrast, in an STR, the cell agglomeration requires collisions between cells to establish cell–cell connections, facilitated by protein-mediated adhesion forces on the cell surface. Cell agglomeration progresses, until the hydrodynamic forces become too high to support further cell attachments to the spheroid. A steady state is reached when the spheroid strength (the total of all cell adhesion forces within the spheroid) equals the hydrodynamic forces in the culture medium.

**Table 1.** Overview of dynamic aggregate/spheroid processes with different cell types, applications, bioreactor types, and the respective scalability in accordance with the similarity theory. We evaluated the production scale, the stirrer type (or similar), and consequently determined the flow range based on own calculations regarding the Reynolds number, Re. Additionally, we listed the seeding density, and the size range of the aggregates and yield per batch. Finally, we compared the described agglomeration techniques, depending on the kind of regulating force, the fluid dynamic, and the ability to adjust the aggregate size. We defined the production of spheroids as controlled when the cells aggregate under isotropic conditions, i.e., at Re > 10,000. This leads to a small size distribution of the aggregates and prevents further agglomeration of the spheroids themselves. Moreover, the applied hydrodynamic forces can restrict the size of the formed spheroids. At ratios of $d_{Sph}/\lambda < 3$, the size restriction is facilitated by surface erosion (for further information, see Supplementary or [30]).

| Cell Type | Cell Name | Application | Bioreactor Type; Scalability | Production Scale [mL] | Stirrer Type | Reynolds Number Re [−] and Flow Range ** | Seeding Density [Cells mL$^{-1}$] | Size Range [μm] | Yield | Agglomeration Technique | Reference |
|---|---|---|---|---|---|---|---|---|---|---|---|
| Islets/ β-cell lines | INS-1 | Bioprocess model for diabetes therapy | STR (Infors HT); scalable | 1000 | 30-SPB; 45-SPB; Rushton | Re = 11,000–18,000; turbulent | $5 \times 10^5$ | 40–50 | $22 \times 10^3$ spheroids mL$^{-1}$; $1.1 \times 10^3$ IEQs mL$^{-1}$; $22 \times 10^6$ spheroids batch$^{-1}$; $1.1 \times 10^6$ IEQs batch$^{-1}$; | Controlled: isotropic conditions; surface erosion ($d_{Sph}/\lambda < 3$); narrow size distribution; size restricted | This study |
|  | MIN6 | Research studies and clinical application | Spinner flask (ProCulture, Corning); semi-scalable | Not specified | Straight blade paddle impeller * | Re = 2300 (d = 40 mm, 60 rpm); non-turbulent | $0.2$–$0.4 \times 10^5$ | 100–400 | ~$0.7 \times 10^6$ cells mL$^{-1}$ | Uncontrolled: non-isotropic; broad size distribution; no size restriction | [26] |
|  | Primary neonatal porcine pancreatic islet cells | Scalable process to expand pancreatic endocrine tissue for cell therapy | Spinner flask (Corning); semi-scalable | 100 | Magnetic stir bars | - | $1.3 \times 10^5$ | Not specified | $1 \times 10^6$ cells mL$^{-1}$; $1 \times 10^8$ cells batch$^{-1}$ | Uncontrolled: non-isotropic; broad size distribution; no size restriction | [25] |
| ESC | CyT49 | Diabetes therapy | 6-well plates; not scalable | 5.5 | No stirrer | Non-turbulent | $10 \times 10^5$ | 100–200 | ~1000 aggregates mL$^{-1}$; ~5500 aggregates batch$^{-1}$ | Uncontrolled: non-isotropic; broad size distribution; no size restriction | [32] |
|  | Royan H5 and H6 | Bioprocess development for production of aggregates | Spinner flask (Cellspin; Integra Biosciences); semi-scalable | 100 | Magnetic pendulums * | - | $2$–$10 \times 10^5$ | 140–200 | $2 \times 10^6$ cells mL$^{-1}$; $2 \times 10^8$ cells batch$^{-1}$ | Uncontrolled: non-isotropic; broad size distribution; no size restriction; addition of shear protectant | [24] |
| iPSC | hCBiPSC2 | Development of suspension culture for iPSCs | STR (cellferm®, DASGIP AG); scalable | 100 | 45° 0°, 45° 60°, 60° 60° impeller | Re ≈ 2500 (d = 40 mm, 60 rpm); Non-turbulent | $4$–$5 \times 10^5$ | 50–150 | $2 \times 10^6$ cells mL$^{-1}$; $2 \times 10^8$ cells batch$^{-1}$ | Uncontrolled: non-isotropic; broad size distribution; no size restriction | [33] |
|  | hHSC_1285i_iPS2 | Development of high-density bioprocessing | STR (DASbox, Eppendorf AG); scalable | 150–500 | Eight-blade impeller (60° pitched) | Re = 1200–2600 (d = 30 mm); non-turbulent | $5 \times 10^5$ | 50–400 | $2.9 \times 10^6$ cells mL$^{-1}$; $4.6$–$14.5 \times 10^8$ cells batch$^{-1}$ | Uncontrolled: non-isotropic; broad size distribution; no size restriction | [34] |
|  | hiPSC1 and hiPSC4 | Bioprocess development for production of iPSCs aggregates | Spinner flask (Cellspin; Integra Biosciences); semi-scalable | 100 | Magnetic pendulums * | - | $2$–$10 \times 10^5$ | 140–200 | $1.8 \times 10^6$ cells mL$^{-1}$; $1.8 \times 10^8$ cells batch$^{-1}$ | Uncontrolled: non-isotropic; broad size distribution; no size restriction; addition of shear protectant | [24] |
| Tumor cells | MCF7, BT474, HCC1954, HCC1806, A549, H460, H157, H1650 and HT29 | 3D cancer models | Baffled spinner flasks (Corning® Life Sciences); semi-scalable | 125–500 | Straight blade paddle impeller | Re = 2300–3800 (d = 40 mm, 60–100 rpm); non-turbulent | $2 \times 10^5$ | 81–298 (depending on cell type) | 1000–1500 spheroids mL$^{-1}$; 0.16–0.63 × 10$^6$ spheroids batch$^{-1}$ | Uncontrolled: non-isotropic; broad size distribution; no size restriction | [27] |

* We assumed the stirrer type and size, based on the product information of the bioreactor system. ** The flow range is based on own calculations regarding the Reynolds number, Re (Re < 10,000 corresponds to non-turbulent and Re > 10,000 to turbulent).

We aim to develop a large-scale STR-based production process for viable and functional β-cell spheroids, allowing the production of large numbers of spheroids within a defined size range of 20 to 100 μm. The term "large-scale" has no precise definition, but we aimed to scale up our process to a working volume of 1 L, which is 100-times higher than the volume of RWVRs, and repeatedly higher than comparable processes with iPSCs (Table 1). We used the INS-1 as our model β-cell line to develop this process. We used shaking flasks as a screening platform to characterize the behavior of INS-1 β-cells in a turbulent flow regime, and to determine the minimal seeding density and spheroid strength. We then used the spheroid strength (equaling the sum of the bonding forces between adjacent cells mediated by membrane proteins [30]) to predict the spheroid size in the STR. Furthermore, we investigated the influence of the stirrer design by testing three stirrer types with different pumping directions (axial, axial/radial and radial), stirrer-swept volumes, and maximum energy dissipation to mean energy dissipation ratios. Finally, we scaled up from the shaking flasks to the STR by keeping the energy dissipation constant.

## 2. Materials and Methods

### 2.1. Cells and Culture Medium

We used the rodent β-cell line INS-1 (kindly provided by Sebastian Hauke from European Molecular Biology Laboratory, Heidelberg, Germany), which originated from a Simian virus-induced rat insulinoma. Pre-cultures, prepared in RPMI 1640 medium (Biochrom, Berlin, Germany) and supplemented with 10% (*v/v*) fetal bovine serum (FBS) and 0.05 M 2-mercaptoethanol (Carl Roth, Karlsruhe, Germany), were incubated at 37 °C in a 5% $CO_2$ atmosphere. The cells were seeded with $5 \times 10^4$ cells cm$^{-2}$ into 25–175 cm$^2$ T-flasks (Sarstedt, Nuembrecht, Germany) and cultured to a confluence of 80–90%. Before passaging, the cells were observed by microscopy to ensure the absence of morphological defects and contamination. The cells were washed once with 0.3 mL cm$^{-2}$ phosphate-buffered saline (PBS, Biochrom) before detachment with 0.012 mL cm$^{-2}$ trypsin (Biochrom) for 5–7 min at 37 °C. After detachment, premature agglomeration was avoided by omitting centrifugation. Although the INS-1 cell line was robust, we did not exceed a passage number of 35.

### 2.2. Static Spheroid Formation

Static 3D cultures as spheroids were generated in 96-well plates (U-bottom) with a cell-repellent surface (Greiner, Kremsmünster, Austria). The INS-1 were seeded at $10^3$ cells per well, where the cells were forced into agglomeration by the parabolic shape. The working volume was 200 μL, and a 50% medium exchange was performed every other day. Daily imaging of the spheroids was used to determine the diameter and circularity, while staining with calcein/ethidium was used to determine the viability. Each experiment was performed in 12-fold biological replicates.

### 2.3. Shaking Flask Cultivation

Preliminary experiments were carried out in shaking flasks at 37 °C in a 5% $CO_2$ atmosphere. We used 100-mL shaking flasks with a working volume of 20 mL. The shaking flasks had an inner diameter of 0.064 m, and four baffles. For all experiments, we used the same Celltron shaking plate (Infors HT, Bottmingen, Switzerland) with an eccentricity of 2.5 cm and varying rotational frequencies. The cells were inoculated with a seeding density of $5 \times 10^5$ cells mL$^{-1}$. Daily samples were stained with calcein/ethidium to test viability, and an image-based analysis of the particle distribution was carried out as described below.

### 2.4. Production of Spheroids in a STR

For the large-scale production of β-cell spheroids, we used the Labfors 5 Bioreactor system (Infors HT). The bioreactor tank had an inner diameter of 0.115 m and a dished bottom. The working volume $V_L$ was 1 L, resulting in a ratio of liquid height to tank diameter of $H_L/D_T$ = 1. We used three different stirrer types: a 30° three-segment pitched-blade

stirrer (30°-3-SPB), a 45° three-segment pitched-blade stirrer (45°-3-SPB), and a Rushton turbine (Table S1). To achieve a constant mean energy dissipation of $\bar{\varepsilon} = 35$ kW kg$^{-1}$ in each STR run, the stirrer speed for the 30°-3-SPB was 183 rpm, for the 45°-3-SPB it was 141 rpm, and for the Rushton turbine it was 162 rpm. The STR was equipped with process analytical technology, including a temperature probe, a pH probe, a dissolved oxygen (DO) probe, and a dielectric spectroscopy probe (all from Hamilton Germany). During cultivation, the temperature was maintained at 37 °C, the pH was regulated by gassing with $CO_2$ or the addition of 1 M NaOH, and the DO concentration was kept above 40% by discontinuous submersed gassing (sparger). A DO of 100% represents oxygen-saturated culture medium (without cells) at 37 °C, the corresponding stirrer speed, and 100 mL min$^{-1}$ continuous submersed gassing with air. Every 12 min, the permittivity (correlating with viable biomass) was measured inline at 1000 kHz, combined with a frequency scan of 300–10,000 kHz. The four installed probes, the shaft guide, the sparger, and the pipes for sampling and harvest were used as baffles, and this was sufficient to achieve complete baffling [35]. After the sterilization of the bioreactor, pre-cultured cells in multiple T-175 flasks were harvested, using trypsin. The cells were in the exponential growth phase and not overgrown. As above, we avoided centrifugation to prevent a premature agglomeration. The seeding density in the STR was $5 \times 10^5$ cells mL$^{-1}$. The cells were stained with calcein/ethidium to test their viability, and an image-based analysis of the particle distribution was carried out as described below.

### 2.5. Calcein/Ethidium Staining to Analyze Particle Count, Size, Circularity, and Areas of Green and Red Particles

We used the LIVE/DEAD Viability/Cytotoxicity Kit (Invitrogen (Waltham, MA, USA), Thermo Fisher Scientific (Waltham, MA, USA)) containing calcein AM and ethidium homodimer 1. Calcein AM is a true live stain that detects intracellular esterase activity, and ethidium homodimer 1 intercalates into DNA to detect dead cells. The cells were stained without washing steps to prevent cell/spheroid loss. After sampling, five technical replicate (100-μL) samples were transferred to flat-bottom 96-well plates. The staining solution was added directly to the sample (final concentration = 2 μM calcein/ethidium) and incubated for 30 min at 37 °C. No washing steps were applied to prevent cell/spheroid loss. The samples were analyzed using a Cytation3 (BioTek Instruments, Winooski, VT, USA). The fluorescent cells and spheroids in entire wells were captured by acquiring multiple images at 10× magnification. The particle count, size, circularity, and areas of green and red particles were determined, using Gen5 v2.07.17 to assess cell viability.

### 2.6. Particle Size Distribution

Image analysis was restricted to the size range 0–300 μm. To distinguish between single cells and spheroids, we set the threshold to 20 μm. All counts <20 μm were defined as single cells, whereas all counts >20 μm were evaluated as spheroids. The size of the particle distribution was expressed using the Sauter diameter $d_{32}$ as shown in Equation (1):

$$d_{32} = 6 \cdot \frac{\sum V_{Sph}}{\sum S_{Sph}} \tag{1}$$

where $S_{Sph}$ represents the surface of the spheroids. We found that the Sauter diameter $d_{32}$ described the size distribution of the spheroids very well compared with the mean diameter, median and modus.

We compared the progression of the single cell and spheroid counts of the culture period in addition to the spheroid distribution (20–300 μm) by constructing box plots, showing the median, mean, minimum (1%) and maximum values (99%). The span width of the spheroid distribution within the margin of 1 to 99% was used to compare the width of the particle distributions from different cultures.

## 2.7. Assessment of Viability

The viability of the whole sample was determined from the respective green $A_{green}$ and red $A_{red}$ areas. Hereby, we discriminated between the areas according to our determined size ranges to determine the viability of the single cells (areas of the size range: 0–20 μm), the spheroids (areas of the size range: 20–300 μm) or the total viability (areas of the size range: 0–300 μm) using the following Equation (2):

$$Via = \frac{A_{green}}{A_{green} + A_{red}} * 100 \tag{2}$$

## 2.8. Determination of Growth Rate and Expansion Factor

The size of the viable spheroids was used to determine the volume growth rate $\mu_{Vol}$ of the spheroids. We assumed that the spherical volume of the spheroids increases due to the exponential cell growth within the spheroids, and that the cells are constant in volume. This leads to the correlation shown in Equation (3):

$$\mu_{Vol} = \frac{ln\left(V_{Sph,2}\right) - ln\left(V_{Sph,1}\right)}{t_2 - t_1} \tag{3}$$

where $V_{Sph}$ corresponds to the spheroid volume at time point $t$.

The resulting volume doubling time $t_{D,Vol}$ can be calculated as shown in Equation (4):

$$t_{D,Vol} = \frac{ln(2)}{\mu_{Vol}} \tag{4}$$

The volume expansion factor $V_{Ex}$ can be expressed as shown in Equation (5):

$$V_{Ex} = \frac{V_{Sph,2}}{V_{Sph,1}} = \frac{d_{Sph,2}^3}{d_{Sph,1}^3} \tag{5}$$

## 2.9. Assessment of β-Cell Functionality

The glucose-dependent insulin secretion of the β-cell spheroids was measured by exposing the cells to varying glucose concentrations, and measuring the secreted insulin. After sampling, three technical replicates of 500 μL were washed twice with PBS to remove the culture medium and possible insulin residues present in FBS. The cells were then incubated in 500 μL medium lacking FBS, but containing 1.1 mM glucose for 40 min at 37 °C. The supernatant was removed, centrifuged, and stored at −20 °C for analysis (sample for basal insulin secretion). We then added 500 μL of medium lacking FBS, but containing 16.7 mM glucose for 20 min at 37 °C. The supernatant was removed, centrifuged, and stored at −20 °C for analysis (sample for acute insulin secretion). The insulin in the supernatants was analyzed, using a rat insulin ELISA kit (DRG Instruments, Marburg, Germany). The samples were measured in duplicate and evaluated within the ELISA working range, using a four-parameter logistic curve.

## 2.10. Statistical Analysis

If not stated otherwise, all experiments were performed as three independent runs, and presented as mean value ± standard deviations (STDV). For statistical analysis, the following were applied: (1) two groups: Student's *t*-test, (2) for more than two groups: a one-way ANOVA followed by a post-hoc analysis using Bonferroni correction. Intervals of significance were indicated as follows: * $p < 0.05$, ** $p < 0.01$, and *** $p < 0.001$.

## 3. Results and Discussion

### 3.1. Differences in Static and Dynamic Formation of INS-1 Spheroids

To gain an initial insight into the differences of the static and the dynamic formation of spheroids, we used cell-repellent 96-well plates as a static system, and baffled shaking flasks as a simple dynamic system, compatible with established models of power consumption [36,37]. Under static conditions, we determined INS-1 agglomerates after 24 h, having $181 \pm 3$ µm in diameter with a circularity of $0.92 \pm 0.05$. Under dynamic conditions, we determined the agglomeration of INS-1 cells within 24 h, producing spheroids with a Sauter diameter $d_{32} = 94 \pm 12$ µm and a circularity of $0.65 \pm 0.02$ (Figure 1). Dynamic and static spheroid formation occurred at similar rates, but the circularity of the static spheroids was significantly (*** $p < 0.001$) higher. The volume growth rate $\mu_{Vol}$ of $0.14 \pm 0.04$ d$^{-1}$ and $t_{D,Vol} = 5.0 \pm 1.4$ d for dynamic spheroid formation was significantly (** $p < 0.01$) lower than the corresponding values for static cultures ($\mu_{Vol} = 0.327 \pm 0.013$ d$^{-1}$, $t_{D,Vol} = 2.12 \pm 0.08$ d). The dynamic growth of spheroids is likely to be restricted by surface erosion, as growing cells in the outer layer are sheared off and/or collide with/adhere to suspended single cells or smaller agglomerates. Spheroids in the size range of 59–269 µm were reported to form after 7 d in shaking cultures of the β-cell line RIN-5F, while maintaining β-cell functionality [38]. The method was not characterized in detail. Based on the information provided, we assume that the cells were not limited by hydrodynamic forces, thus allowing a continuous spheroid growth. Our static spheroids developed a necrotic core and dropped to ~50% viability after 24 h due to mass transfer limitations, whereas the viability of spheroids formed in a dynamic culture was always close to 100% (except some single cells within the spheroids) regardless of their size (up to 200 µm). Similarly, agglomerates of the rodent β-cell line MIN6 were produced after 2 d in static culture dishes and spinner flasks. Although the spheroid concentration remained constant while they grew from 100 to 400 µm, a dense necrotic core formed in the static spheroids (viability ~65%), whereas those in the spinner flasks maintained ~85% viability [26]. The spheroids in the dynamic culture were also characterized by a lower lactate dehydrogenase and caspase activity, and lower detected levels of fragmented DNA, using the TUNEL assay [26].

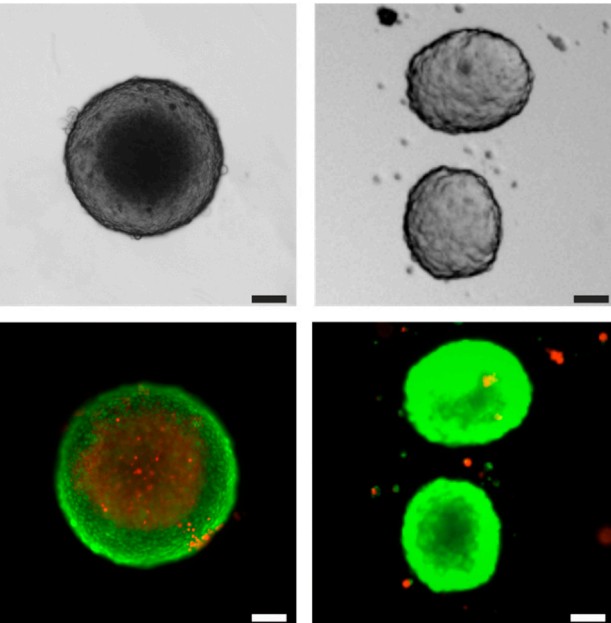

**Figure 1.** Comparison of spheroid structures. Left panels show INS-1 spheroids produced in static 96-well plates with cell-repellent surfaces. These spheroids developed a necrotic core, seen as a denser area in the bright field image (**upper**), and the corresponding red area by live/dead staining (**lower**). Right panels show highly viable spheroids produced in dynamic shaking flask cultures. Scale bar = 100 µm.

*3.2. Influence of Seeding Density on INS-1 Spheroid Formation in Shaking Flasks*

Next, we investigated relevant process parameters such as seeding density, turbulence working range, and spheroid strength in shaking flasks. We tested seeding densities from $1 \times 10^4$ to $1 \times 10^6$ cells mL$^{-1}$. At values between $1 \times 10^5$ and $6 \times 10^5$ cells mL$^{-1}$, we observed a plateau of a spheroid size with a narrow size distribution ($d_{32} = 63 \pm 5$ µm) and a rapid spheroid formation (within 24 h) and, therefore, defined $1 \times 10^5$ to $6 \times 10^5$ cells mL$^{-1}$ as our working range. At values below $1 \times 10^5$ cells mL$^{-1}$, agglomeration was inhibited ($d_{32} = 29 \pm 7$ µm, ** $p < 0.01$), whereas values above $6 \times 10^5$ cells mL$^{-1}$ generated larger spheroids ($d_{32} = 82 \pm 3$ µm, ** $p < 0.01$) due to further visible agglomeration of the spheroids themselves. The seeding density is not only important for a dynamic spheroid formation, but also affects the functionality of the β-cells, which is dependent on the cell number in each spheroid. For example, when spinner flasks were used to form spheroids from primary neonatal porcine pancreatic islet cells at different seeding densities, low values ($6.3 \times 10^3$ and $5 \times 10^4$ cells mL$^{-1}$) reduced the number of aggregates, whereas the highest value ($1.3 \times 10^5$ cells mL$^{-1}$) increased the number of islet-like aggregates containing insulin-positive cells [25]. The positive effect reflected the increase in cell–cell interactions at higher seeding densities. Our chosen working range also matches the values reported for other cell types that grow as spheroids (Table 1), including iPSCs [39,40]. We therefore carried out all subsequent experiments with a seeding density of $5 \times 10^5$ cells mL$^{-1}$.

*3.3. Influence of Power Input on INS-1 Spheroid Formation in Shaking Flasks*

The development of a turbulent flow regime is necessary for the scale-up of the INS-1 spheroid formation process in a STR. Based on earlier models [36,37], we calculated a Reynolds number of Re = $10^4$ (fully developed turbulence) in the shaking flasks at a frequency of 100 rpm. We investigated the spheroid formation at shaking frequencies in the range between 90–130 rpm, and correlated the spheroid size to the power input. We observed a linear decrease in the spheroid size between 100–120 rpm (corresponding to 35–60 W m$^{-3}$), combined with a narrow span width of the spheroid distribution of $85 \pm 28$ µm, where we anticipated the turbulent flow regime. Spheroids produced with an increased power input (and thereby increasing hydrodynamic forces) showed the anticipated and significant (** $p < 0.01$) decrease in size from $d_{32} = 86 \pm 6$ µm at 100 rpm to $d_{32} = 44 \pm 5$ µm at 120 rpm (Figure 2). An increasing stirrer speed also reduced the size of hPSCs spheroids [34] and, similar to our results, the viability of the spheroids remained high, although increasing shear forces led to a surface erosion and the loss of cells from the spheroid surface. Therefore, we determined an increasing amount of dead single cells with an increasing power input (35–60 W m$^{-3}$). In relation to 100 rpm, spheroids produced under non-turbulent conditions (90 rpm, 28 W m$^{-3}$) were twice as large on average ($d_{32} = 168 \pm 4$ µm, ** $p < 0.01$) and were distributed over a broader size range (span width: $232 \pm 24$ µm), whereas no spheroids were formed at 130 rpm (=74 W m$^{-3}$) and the size distribution represented the profile of a single-cell suspension ($d_{32} = 10 \pm 3$ µm, ** $p < 0.01$). Moreover, the efficiency of spheroid formation, represented as the number of formed spheroids in relation to the total cell number (Figure 2), fell to low values of 6% and 5% at 90 and 130 rpm, respectively. Within the turbulent range (100–120 rpm), the spheroid formation efficiency increased to 27% (110 rpm). These data agreed with the spheroid formation theory discussed in the supplementary, which we adapted from particle systems such as clay, latex, and glass [41–46]. Based on the ratio $d_{Sph}/\lambda$, we concluded that the INS-1 spheroid formation process reached a steady-state diameter that reflected the shear forces of the corresponding eddies acting on the spheroid surface. This assumption is supported by the decreasing spheroid size as the power input increases. The spheroid size is, therefore, limited by the Kolmogorov eddy size, which can also be described as the equilibrium between kinetic energy in the culture medium and the bonding energy of the spheroids.

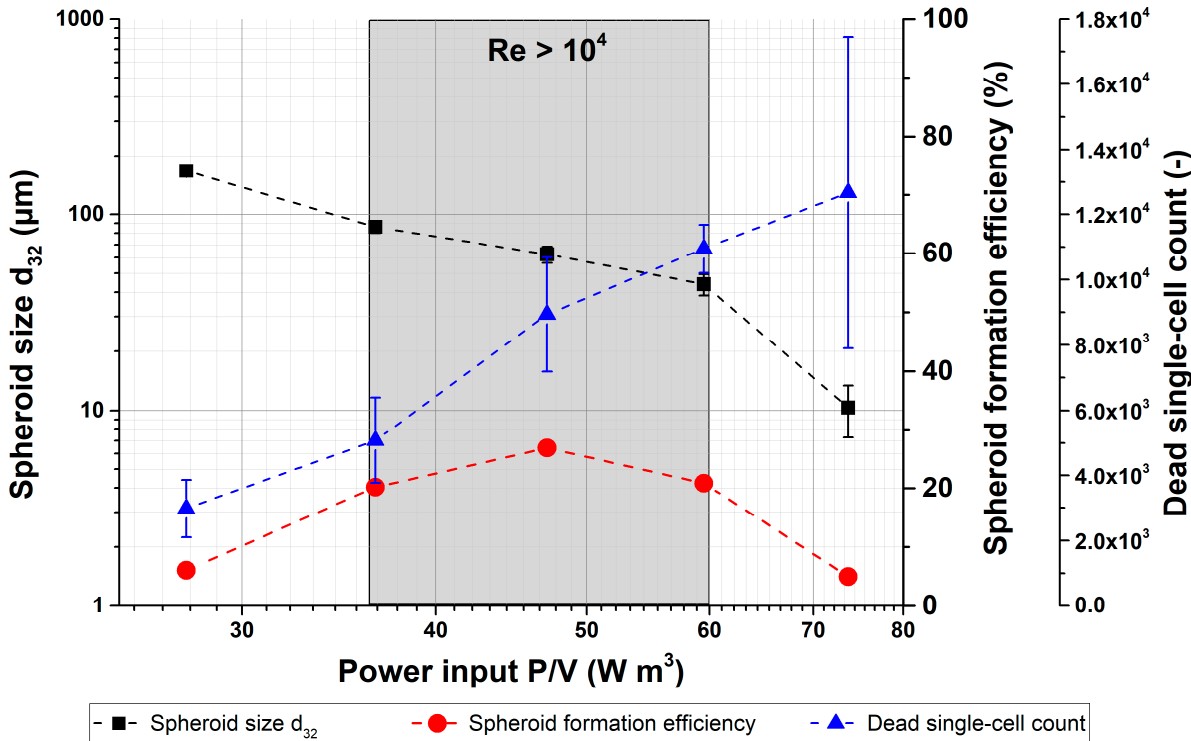

**Figure 2.** Spheroid size ($d_{32}$, black squares) from dynamic cultivation in shaking flasks ($n = 3$, error bars represent STDV). Spheroids produced with a power input lower than the gray highlighted working range (35–60 W m$^{-3}$; 100–120 rpm) were twice as large, whereas increasing shear forces reduced the steady-state size until agglomeration was prevented at a P/V of $\approx$75 W m$^{-3}$ (130 rpm). An increase in power input displaced cells from the spheroid surface, leading to greater numbers of suspended dead singe-cells (blue triangles) in the supernatant. The spheroid formation efficiency (the number of formed spheroids in relation to the total cell number; red dots) supported the turbulent range from 100 to 120 rpm with an increased efficiency above 20%.

*3.4. Determination of INS-1 Spheroid Strength in Shaking Flasks*

The spheroid strength is an essential parameter in our process, so we estimated the value in shaking flask experiments. The spheroid strength is influenced by the strength and number of bonds (F$_{ad}$) between cells, the size and shape of the cells, and the compaction of the spheroid. The particle analysis and wastewater treatment literature provide multiple approaches to determine spheroid strength [41,44,47,48], which we adapted for the INS-1 spheroids. These methods are based on the relationship between the power input/energy dissipation and the corresponding floc/spheroid diameter, thus distinguishing between agglomerate splitting by tensile forces, and surface erosion due to shear forces. However, we are aware that the agglomeration of non-biological particle systems is enabled by polymers or electrostatic interactions, whereas the agglomeration of cells is also based on the interaction between surface proteins (such as integrins and cadherins) on adjacent cells [49]. This is a highly variable precondition for different cell types and even for the same cell type, because the expression of surface proteins on cells can change during growth, senescence, and differentiation, and can be influenced by cultivation and harvesting methods.

Because there is no standardized procedure to evaluate spheroid strength, we produced spheroids within the turbulent working range under increasing power input, and correlated the steady-state spheroid size (after 24 h) with energy dissipation (Figure 2). We applied the following method [41,50] to describe the steady-state diameter *d* using Equation (6):

$$d = C_{Agg} \cdot G^{-\gamma} \tag{6}$$

where $C_{Agg}$ is the strength of the agglomerate (here, spheroid strength), $\gamma$ is the stable agglomerate size exponent, and the shear rate $G$ is defined using Equation (7):

$$G = \sqrt{\frac{\epsilon}{\vartheta_L}} \qquad (7)$$

We calculated for the exponent $\gamma$ a value of 2.8 ($R^2 = 0.99$) over the linear region of the equation. The exponent $\gamma$ provides insight into the agglomeration behavior of the INS-1 spheroids: $\gamma$ values of ~0.5 indicate fragmentation by tensile forces, whereas $\gamma > 2$ suggests surface erosion [50]. This fits to our data shown above that the increasing amount of dead single cells with increasing power input (35–60 W m$^{-3}$) is the result of surface erosion and the loss of cells from the spheroid surface.

We finally calculated a strength of $325 \pm 5$ N m$^2$ for the INS-1 spheroids. Our spheroid strength was similar in magnitude to monolayer cells interacting with planar surfaces. Here, the cells also attached to the surface via surface proteins such as integrins, and the sum of $F_{ad}$ corresponded to the attachment strength. A single fibroblast displaying ~$2 \times 10^5$ integrins on its surface, required a force of 400 N m$^2$ to detach the cell from the surface [51]. Notably, the spheroid strength we calculated was only valid for the first 24 h, and probably increased during cultivation due to the buildup of ECM components, and the rearrangement of cells led to spheroid compaction. The spheroid strength we determined was also only valid for the INS-1 under our specific experimental set-up, as the spheroid strength is highly dependent on the cell type and culture conditions. For example, we observed completely different spheroid strengths when comparing β-cells cultivated in a serum-free and in a serum-supplemented medium.

### 3.5. INS-1 Spheroid Formation in a Stirred-Tank Bioreactor Using Different Stirrer Types

Our investigations in shaking flasks served as a basis for the 1 L scale production of INS-1 spheroids in a fully controlled and monitored STR. We therefore chose the power input/mean energy dissipation from the shaking flask experiments (35 W m$^{-3}$), where we achieved a good INS-1 spheroid formation and kept that value constant in the STR process. We then investigated the influence of three different stirrer types. Each stirrer has a specific power number $N_P$ that reflects the momentum of resistance, and, therefore, the reinforcement of the power input into the culture medium. Although higher $N_P$ values are often associated with greater particle stress, several investigations have shown that the same power input for an axial flow stirrer with a lower $N_P$ can cause more particle disintegration [52–55]. This may reflect the so-called energy dissipation circulation function (EDCF), which describes the particle stress in relation to the particle residence time and frequency within the stirrer-swept volume $V_S$, where the greatest particle stress occurs. Logically, axial pumping stirrers with low $N_P$ values must increase the stirrer frequency to achieve the same power input, and this increases particle stress. Furthermore, the trailing vortex behind the stirrer blade is the region with $\varepsilon_{max}$, and is thus responsible for the destructive effects [54]. To investigate these effects, we selected three different stirrer types, varying in $N_P$ and $V_S$, based on axial, axial/radial and radial pumping orders (Table S1).

We ran each stirrer type with the same power input/mean energy dissipation. After 24 h, we recorded values of $d_{32} = 94 \pm 12$ μm for the shaking flask (reference) cultures, $d_{32} = 51.5 \pm 1.2$ μm for the 30°-3-SPB stirrer, $\underline{d}_{32} = 40 \pm 3$ μm for the 45°-3-SPB stirrer, and $d_{32} = 50.6 \pm 1.6$ μm for the Rushton turbine (Table 2). The spheroid size was significantly (*** $p < 0.001$) decreased in the STR, while the 30°-3-SPB and Rushton turbine produced a similar spheroid size. Spheroids generated with the 45°-3-SPB were significantly (* $p < 0.05$) smaller. Although shaking flasks achieve a homogenous energy dissipation, whereas energy dissipation in the STR is heterogeneous due to the power input of the centrally-installed stirrer, the higher value for $d_{32}$ and the larger span width of the size distribution (Table 2) may reflect the manual manufacturing process and variations in shaking flask geometry. Although we aimed to transfer the energy dissipation from the shaking flasks to the STR to reproduce the results of the shaking flask experiments, we anticipated differences in the

steady-state size of the spheroids after 24 h. It was challenging to measure the power input in baffled shaking flasks due to variations in shape, size, and number of baffles, hence the corresponding models may overestimate the energy dissipation. The overestimation of hydrodynamic forces was supported by the significantly (*** $p < 0.001$) larger $d_{32}$ and a lower and decreasing single-cell count, given there is less stress on the spheroid surface. Furthermore, there were substantial differences in power input between the shaking flasks and STR. The power input in shaking flasks was reinforced by friction between the culture medium and the vessel wall, and the redirection of the tangential flow into radial flow by the baffles, whereas the power input in STRs was generated by the stirrer. Even so, we found that shaking flasks were suitable for preliminary experiments to guide the manufacture of spheroids, if turbulent conditions were used to generate homogenous stress.

**Table 2.** Overview of relevant stirrer properties such as the power number $N_P$, energy dissipation circulation function (EDCF), stirrer tip speed, and the ratio of maximum energy dissipation to the mean $\varepsilon_{max}/\bar{\varepsilon}$ (calculated by [56]). Further, we summarized important particle parameters such as the Sauter diameter $d_{32}$, the spheroid formation efficiency, and the span width of the spheroid distribution from 1 to 99% after 24 h.

| Stirrer Type | Np [−] | $d_{32}$ [µm] | Spheroid Formation Efficiency [%] | Span Width of Distribution [µm] | EDCF [kW m$^{-3}$ s$^{-1}$] | Stirrer Tip Speed [m s$^{-1}$] | $\varepsilon_{max}/\bar{\varepsilon}$ [−] |
|---|---|---|---|---|---|---|---|
| Shaking flask | - | $94 \pm 12$ | $7.7 \pm 1.4$ | $134 \pm 11$ | - | - | - |
| 30°-3-SPB | 1.1 | $51.5 \pm 1.2$ | $60 \pm 3$ | $66 \pm 7$ | 0.9 | 0.62 | 5 |
| 45°-3-SPB | 2.4 | $40 \pm 3$ | $8.7 \pm 1.4$ | $51 \pm 6$ | 0.6 | 0.49 | 6 |
| Rushton | 4.0 | $50.6 \pm 1.6$ | $33 \pm 2$ | $61 \pm 3$ | 1.6 | 0.46 | 13 |

The three stirrer types are compared in Figure 3, which shows the counts for single cells (0–20 µm) and spheroids, in addition to the spheroid size distribution as box plots. The desirable outcome would be low single-cell counts and a high spheroid count, while maintaining a narrow spheroid size distribution. The 30°-3-SPB stirrer achieved a continuous low single-cell count (maximum 60,000 cells mL$^{-1}$), indicating that most of the cells formed spheroids, which is also reflected by the high spheroid formation efficiency of $60 \pm 3$% after 1 d (Table 2). The increasing $d_{32}$, while maintaining a narrow span width of the distribution of $66 \pm 7$ µm on day 1 to $100 \pm 6$ µm on day 4, confirmed efficient spheroid growth with low surface erosion, and therefore, few single cells in suspension. In contrast, the 45°-3-SPB stirrer was associated with a high single-cell count (up to 150,000 cells mL$^{-1}$) throughout the cultivation period, while maintaining a similar span width from $51 \pm 5$ µm on day 1 to $79 \pm 21$ µm on day 4. The low spheroid formation efficiency of ~10% and the high single-cell count indicated destructive hydrodynamic forces. The Rushton turbine performance was midway between the other stirrers. The highest $d_{32}$ was achieved after 24 h, and the hydrodynamic forces prevented further spheroid growth and even reduced the spheroid size while increasing the single-cell count. The span width of the spheroid distribution ranged from $61 \pm 2$ µm on day 1 to $88 \pm 3$ µm on day 4. The spheroid formation efficiency was ~35%. Although the increased single-cell count suggested a negative effect, it was a consequence of the surface erosion, and this confirmed our adapted model showing that spheroid formation and size are restricted by surface erosion, as reported for other particles [41]. Although we observed increasing single-cell counts for all three stirrers over time, spheroid growth was not limited in the STR with the 30°-3-SPB stirrer. The overall low single-cell count and high agglomeration efficiency of 60% indicate that the stress caused by hydrodynamic forces was acceptable with this device, in contrast to the size-limiting surface erosion resulting in higher single-cell counts with the 45°-3-SPB stirrer and Rushton turbine.

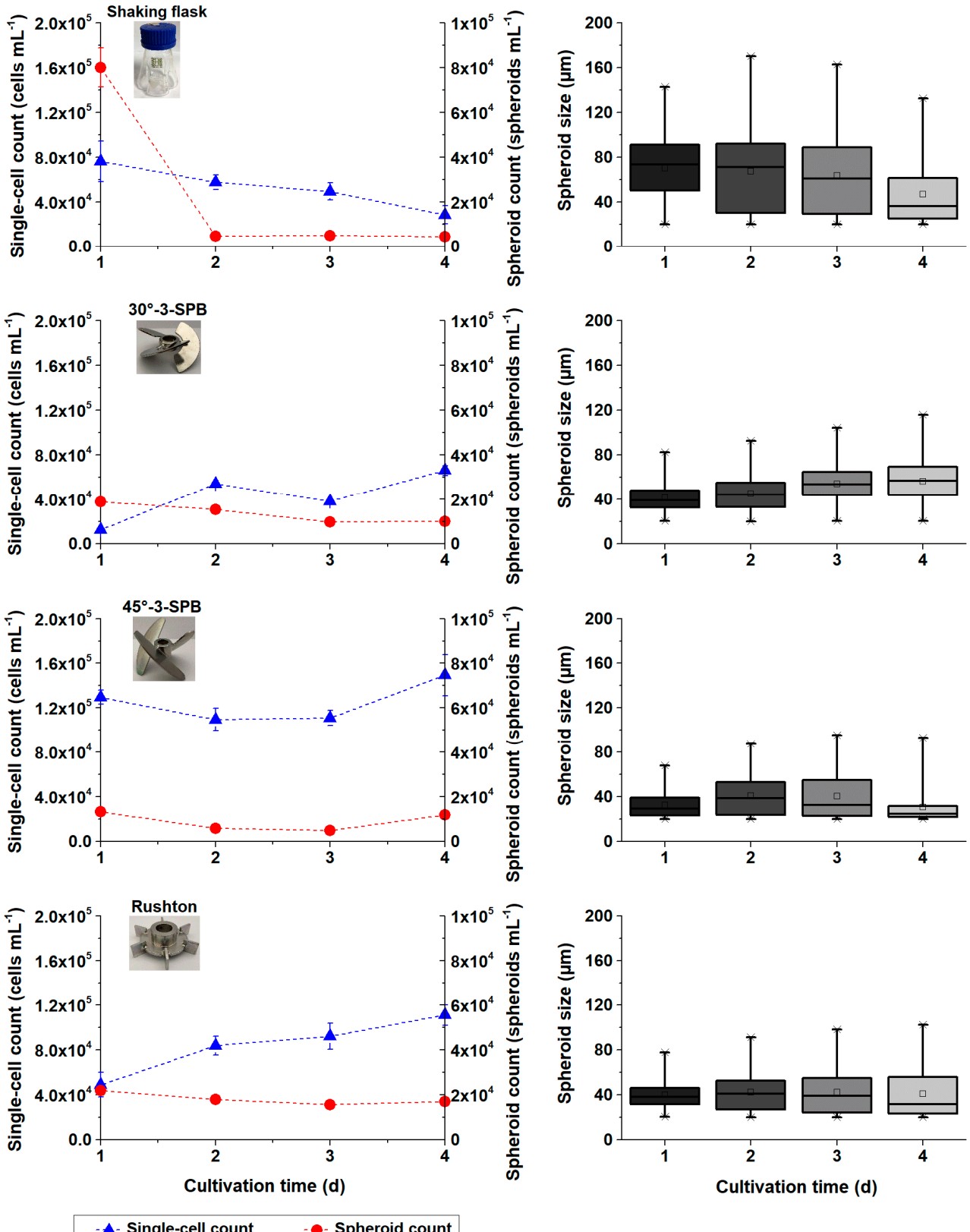

**Figure 3.** Particle counts for the shaking flask cultures (*n* = 3) and the three different stirrer types (*n* = 2) during cultivation for 4 d. Left column: single-cell count (blue triangles, size range: 0–20 μm, error bars = STDV) and spheroid count (red circles, size range: 20–300 μm, error bars = STDV). Right column: box plots of the spheroid size distribution showing the median (straight line), mean (square), minimum value 1% (due to our threshold at 20 μm), and maximum value 99%, each as cross.

Given the power numbers $N_P$ of the three stirrers, our results support the statement that low $N_P$ values (30°-3-SPB) cause less particle stress and encourage spheroid growth, whereas high $N_P$ values (Rushton) produce smaller spheroids. This was also confirmed by the higher EDCF of the Rushton turbine (1.6 kW m$^{-3}$ s$^{-1}$), compared with the lower value for the 30°-3-SPB stirrer (0.9 kW m$^{-3}$ s$^{-1}$). The 45°-3-SPB stirrer had the lowest EDCF (0.6 kW m$^{-3}$ s$^{-1}$), reflecting the 33% increase in vs. and, thus, the lowest stirrer frequency (Table S1). Although the stirrer tip speed is often used to describe the hydrodynamic forces during cell cultivation, in our case the stirrer tip speed provided an insufficient correlation. Whereas the Rushton turbine tip speed was 0.46 m s$^{-1}$, that of the 30°-3-SPB stirrer was 26% higher (0.62 m s$^{-1}$) and should have generated higher shear forces, which disagrees with our observations (Table 2). The 30°-3-SPB stirrer showed a slightly lower $\varepsilon_{max}/\bar{\varepsilon}$ ratio of 5 (assuming a more homogenous energy dissipation in the STR) than the 45°-3-SPB stirrer with a value of 6 (Table 2), which may help to explain the latter's poor performance, although this was nevertheless unexpected. An evaluation of multiple axial flowing stirrers, used to suspend microcarriers for the expansion of anchorage-dependent cells under low shear-stress conditions, resulted in the selection of an elephant ear impeller with a similar design to our 45°-3-SPB device [57]. The large vs. was found to facilitate the establishment of a microcarrier suspension while maintaining low shear rates and collisions between microcarriers (expressed as the turbulent collision severity) compared with axial stirrers with lower vs. and smaller blades to reinforce the power input (such as pitched-blade turbines). The reduced collision rate of this stirrer type could explain the poor performance of the 45°-3-SPB. Further, the angle of the stirrer blades, which affects the suspension efficiency, may also have contributed to our observed results. The 45°-3-SPB stirrer uses different mechanisms to suspend large and fine particles [58] and this could affect the agglomeration process, hence the high single-cell count associated with this stirrer from the start of the process.

### 3.6. Correlation between Predicted and Measured INS-1 Spheroid Size in STR

To correlate our experimental data with our theoretical background of the spheroid formation, we calculated the spheroid size in the STR after 24 h using our determined spheroid strength $C_{Agg}$ (from the shaking flasks experiments) in Equation (8) [43]:

$$d = C_{Agg}^{l} \cdot d_{cell}^{m} \cdot \varepsilon^{-n} \tag{8}$$

where $d_{cell}$ is the single-cell diameter, $\varepsilon$ is the mean energy dissipation, and the exponents are $l = 0.5$, $m = 0$ (in the case of surface erosion) and $n = 0.25$.

Figure 4 shows the calculated spheroid size for each stirrer type in relation to the power input. For the calculations, we assumed that the spheroid strength is constant during the initial 24-h agglomeration phase. Our calculation of the spheroid size in the STR under different power inputs resulted in a good fit for the 45°-3-SPB stirrer, but for the 30°-3-SPB stirrer and Rushton turbine our calculation was off by ~10 μm. This deviation may reflect the different stirrer designs, because the 30°-3-SPB and the Rushton devices showed a similar behavior in terms of particle distribution (reduced single-cell count and higher spheroid formation efficiency) in contrast to the 45°-3-SPB stirrer. Although macroscale eddies are directly affected by the stirrer design and setup (e.g., $d_S/D_T$ and baffling), eddies in the dissipation range are dependent on the fluid viscosity. However, the design and setup of the stirrer influence the magnitude of the energy level of $\lambda$, thus the sensitivity of our model is dependent on the precise characterization and perhaps adaption of the power consumption of the STR/stirrer combination. The same applies for the determination of $C_{Agg}$. Furthermore, the initial agglomeration process may differ for each stirrer type, and the 45°-3-SPB device could promote the formation of smaller, denser spheroids and, consequently, a different $C_{Agg}$. Our model, therefore, requires further empirical verification and assessment for robustness. We must still verify that spheroids are really affected by surface erosion (and not by tensile forces) and that similar conditions are found in the STR, to allow precise measurement of the effect.

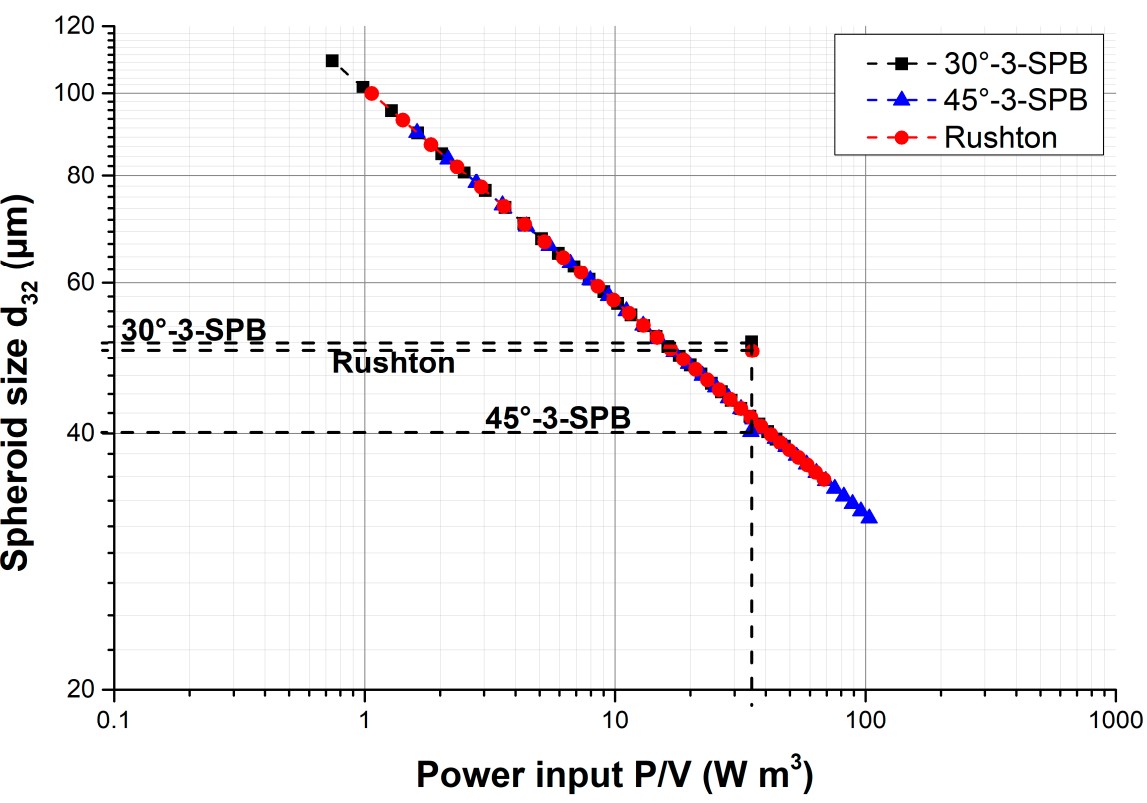

**Figure 4.** The relationship between spheroid size and power input. We determined the spheroid strength ($C_{Agg}$) in shaking flasks to predict the spheroid size in the STR at different power inputs (P/V) after 24 h. The relationship for each stirrer type shows the anticipated characteristic of a declining spheroid size with increasing P/V. With a constant P/V of 35 W m$^{-3}$ (vertical dashed line), the measured spheroid size (horizontal dashed lines) for the 45°-3-SPB stirrer (blue triangle) fit the predicted spheroid size, whereas the 30°-3-SPB stirrer (black square) and the Rushton turbine (red circle) were offset by ~10 μm from the predicted value.

### 3.7. Growth and Viability of the INS-1 Spheroids Produced in the STR

The three different stirrer types resulted in the production of INS-1 spheroids, but the key objective was to produce viable and functional β-cell spheroids. We therefore evaluated the influence of each stirrer type on spheroid growth and viability for a further insight into INS-1 behavior under dynamic conditions. As described for the shaking flasks, we observed volume expansion either due to the attachment of single cells to the spheroids or cell growth within them. We calculated similar volume-based growth rates $\mu_{Vol}$ and volume doubling times $t_{D,Vol}$ for each stirrer type: 30°-3-SPB $\mu_{Vol} = 0.38 \pm 0.018$ d$^{-1}$ ($t_{D,Vol} = 1.81 \pm 0.09$ d); 45°-3-SPB $\mu_{Vol} = 0.4 \pm 0.2$ d$^{-1}$ ($t_{D,Vol} = 1.9 \pm 1.0$ d); and Rushton $\mu_{Vol} = 0.27 \pm 0.072$ d$^{-1}$ ($t_{D,Vol} = 2.7 \pm 0.7$ d) (Table 3). The volume expansion of the spheroids depends on the culture system and the connected hydrodynamic conditions, as [59] could show that the fold expansion of hPSCs spheroids was highest in spinner flasks, compared with well plates and a ring-shaped culture vessel. Interestingly, the low fold expansion of the ring-shaped culture vessel was associated with a higher cell injury, which is attributed to increased shear forces. We hypothesize that this spheroid growth mainly reflected internal cell growth or an increase in spheroid strength due to compaction and stronger adhesion, thus, the extended buildup of an ECM. If we consider cell division within the spheroids, there should be no increase in size when one cell divides into two daughter cells, but if both daughter cells increase their volume or/and become surrounded by an extended ECM, the spheroid should expand. Other studies conclude an increase in spheroid size by cell divisions within the spheroids [26,34]. If the cells in the spheroid are protected by the ECM, the shear forces acting on the surface would need to be stronger than the force required to dislodge single

cells that have adhered to the surface. Therefore, the viability and the count of single cells in the supernatant gives additional information about the performance of each stirrer type (Figure 5). The viability of the spheroids remained close to 100%, whereas we determined an accumulation of dead single cells over the cultivation period. Here, using the 30°-3-SPB stirrer, the continuous low single-cell count during the culture period was also connected to a high total viability (equaling single-cell viability and spheroid viability combined) of 90%. In contrast, high single-cell counts produced by the 45°-3-SPB were connected to low total viabilities down to 58%. In coherence with increasing single-cell counts, the Rushton turbine started with a high total viability of 91%, but revealed a reduction in total viability to 62% after day 2. In agreement with most of the dynamic cultivations, the increased mass transfer prevented the development of a necrotic spheroid core and resulted in an overall high viability of cells within the spheroids [26,33,34,59,60].

**Table 3.** Overview of relevant biological properties of the spheroids produced with each stirrer type, such as the volume-based growth rate $\mu_{Vol}$, and the corresponding minimal time for doubling the volume $t_{D,Vol}$, the acute insulin secretion, and the insulin stimulation index SI (except for the 45°-3-SPB, due to poor performance). Further, we added the yield of spheroids $Y_{Spheroids}$, IEQs $Y_{IEQs}$, and cells $Y_{cells}$ for each 1-L process.

| Stirrer Type | $\mu_{Vol}$ [d$^{-1}$] | $t_{D,Vol}$ [d] | Acute Insulin Secretion [µg h$^{-1}$ L$^{-1}$] | SI [−] | $Y_{Spheroids}$ [Spheroids L$^{-1}$] | $Y_{IEQs}$ [IEQs L$^{-1}$] | $Y_{cells}$ [Cells L$^{-1}$] |
|---|---|---|---|---|---|---|---|
| **30°-3-SPB** | 0.38 ± 0.018 | 1.81 ± 0.09 | 32 ± 3 | 3.2 ± 1.2 | $18 \times 10^6 \pm 2 \times 10^6$ | $1.4 \times 10^6 \pm 0.6 \times 10^6$ | $2 \times 10^8 \pm 0.1 \times 10^8$ |
| **45°-3-SPB** | 0.4 ± 0.2 | 1.9 ± 1.0 | - | - | $14 \times 10^6 \pm 4 \times 10^6$ | $0.3 \times 10^6 \pm 0.2 \times 10^6$ | $0.8 \times 10^8 \pm 0.4 \times 10^8$ |
| **Rushton** | 0.27 ± 0.07 | 2.7 ± 0.7 | 17 ± 4 | 2.7 ± 1.0 | $22 \times 10^6 \pm 0.6 \times 10^6$ | $1.1 \times 10^6 \pm 0.4 \times 10^6$ | $2.2 \times 10^8 \pm 0.2 \times 10^8$ |

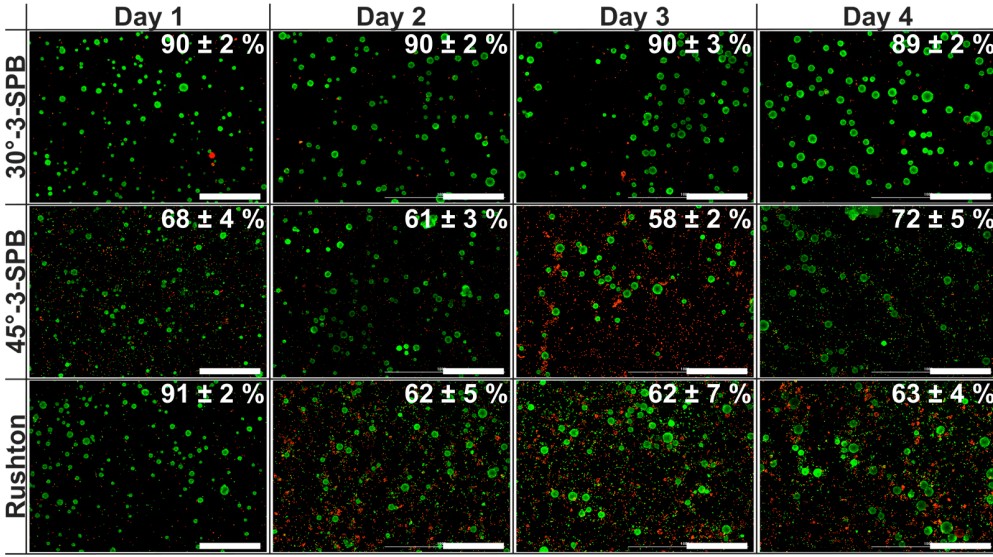

**Figure 5.** Analysis of the total viability of the STR process with each stirrer type over the culture period of 4 d (*n* = 2). Live/dead staining revealed high viabilities for the spheroids, whereas increasing amounts of dead single cells decreased the total viability, shown as mean ± STDV. The scale bar represents 500 µm.

We used the STR and stirrer-associated forces to regulate the size of the INS-1 spheroids, whereas others perform the contrary by using a rotational microgravity cell culture system to minimize any mechanical forces, thereby prolonging the shelf-life of primary β-cells [28]. As we conducted our proof-of concept with a β-cell line, we must consider that islet-derived β-cells are more sensitive to environmental changes and the process conditions. Islets cultivated with simulated microgravity maintained their structural integrity, were more potent, and functional for a longer time, whereas islets under static conditions developed necrotic cores, lost their exocrine mantle, and started to disintegrate. This reflected the dependence

of β-cells on a sufficient diffusive supply of nutrients to maintain their complex microenvironment and functionality. The total viability of the STR process could be maintained (30°-3-SPB) or decreased (45°-3-SPB and Rushton turbine) over time due to surface erosion, but effective mixing in the STR increased the mass transfer, maintaining the viability of the spheroids close to 100%, despite the major impact of hydrodynamic forces (Figure 5).

### 3.8. Online Monitoring of Spheroid Growth and Destruction

The fact that with increasing power input, the spheroids were attacked by surface erosion, was supported by our inline dielectric spectroscopy data (Figure 6). The dielectric spectroscopy probe only detects the viable biomass (dead cells cannot be polarized) and does not distinguish between single cells and cells incorporated into spheroids, which was why we also measured spheroid size offline. Figure 6 shows the biomass signal relative to the offline data. We cultured the INS-1 cells with a low volumetric power input of 5.5 W m$^{-3}$ (75 rpm) for 4 d, and then increased the P/V stepwise to up to 104 W m$^{-3}$ (200 rpm) to induce spheroid destruction. Until day 4, we observed the exponential growth of INS-1 cells, which was demonstrated by the rapidly increasing spheroid size (up to $d_{32} = 182 \pm 9$ µm) and the increasing biomass signal. More importantly, increasing the stirrer speed to 200 rpm reduced the spheroid size and inline biomass signal, and increased the single-cell counts, while maintaining high spheroid viability. This indicated that shear stress, acting on the spheroid surface due to surface erosion ($d_{Sph}/\lambda < 3$), and causing the disruption of ECM-embedded cells, leads to permanent cell damage, as evidenced by the declining biomass signal. The dielectric spectroscopy was also implemented for the expansion and hepatic differentiation of hiPSC spheroids, and showed a good correlation between offline data and online biomass [61].

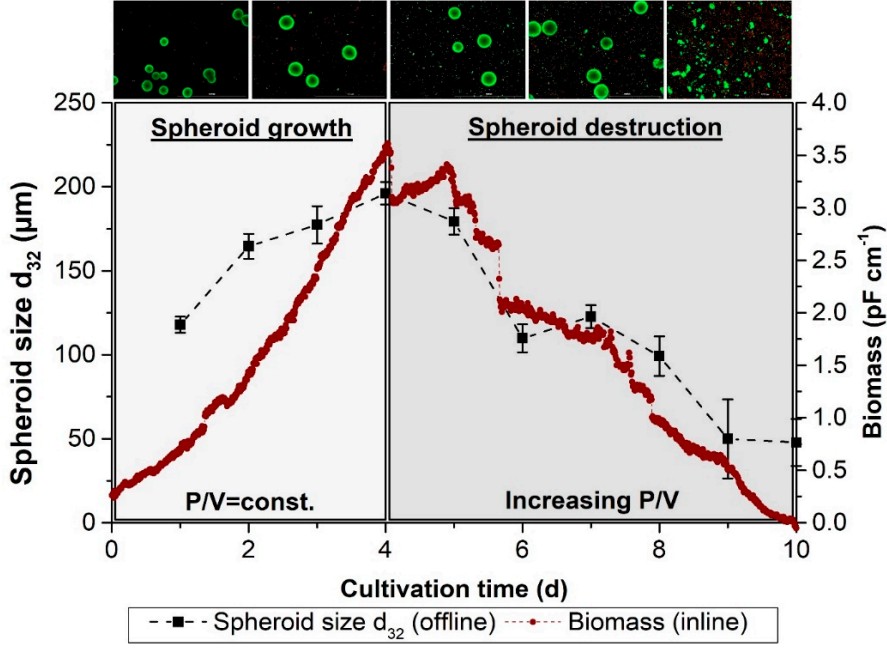

**Figure 6.** Representative spheroid production in an STR with the 45°-3-SPB stirrer. Dielectric spectroscopy showed an increase in biomass (red line) during the first 4 d of cultivation with a constant power input per volume (P/V). As dielectric spectroscopy only measures the volume of viable cells and does not distinguish between suspended individual cells and cells within spheroids, we also plotted the spheroid growth measured offline (black squares and dashed line; mean of 5 technical replicates, error bars = STDV). Upper lane: live/dead images of progressing spheroid growth and destruction. Increasing the P/V led to the disruption of the spheroids, resulting in single cells and smaller agglomerates, which were permanently affected by the hydrodynamic forces, clearly indicated by the decreasing biomass signal and the offline measurement.

### 3.9. Glucose-Stimulated Insulin Secretion

For the direct analysis of functionality, we tested the glucose-stimulated insulin secretion of the INS-1 spheroids. The responsiveness of β-cells to varying glucose levels provides more insight into the health of the cells, than the testing of the basal insulin secretion only. Although we observed wide deviations within the samples, reflecting the manual sample preparation method and variation among the spheroids in each sample, the INS-1 spheroids from all STR runs achieved a higher insulin secretion during the acute phase compared with basal secretion (Figure 7). These results supported the particle analysis and viability data. The 30°-3-SPB stirrer and the Rushton turbine promoted an efficient insulin secretion: $17 \pm 4$ μg h$^{-1}$ L$^{-1}$ with a stimulation index (SI) of $2.7 \pm 1.0$, and $32 \pm 3$ μg h$^{-1}$ L$^{-1}$ with a SI of $3.2 \pm 1.2$, respectively. The 45°-3-SPB stirrer promoted low levels of insulin secretion: $0.8 \pm 0.3$ μg h$^{-1}$ L$^{-1}$ (SI = $13 \pm 7$, solely for completeness, not significant). The poor performance of the 45°-3-SPB stirrer aligned with the accumulation of dead single cells (Figure 3), providing evidence of excessive stress. For the insulin secretion rate per spheroid, the 30°-3-SPB stirrer and the Rushton turbine achieved similar rates of $12 \pm 6$ and $12 \pm 2$ pg h$^{-1}$ spheroid$^{-1}$, respectively, compared with the low value of $0.4 \pm 0.5$ pg h$^{-1}$ spheroid$^{-1}$ for the 45°-3-SPB device. We used an empirically determined conversion factor to approximate the number of cells contained in a spheroid and, thus, compensated for differences in spheroid size for each stirrer type, as previously recommended (Table 3) [21,62]. The Rushton turbine achieved the highest level of insulin secretion per $10^6$ cells ($97 \pm 46$ ng (h $10^6$ cells)$^{-1}$), followed by the 30°-3-SPB stirrer ($69 \pm 43$ ng (h $10^6$ cells)$^{-1}$), and finally the 45°-3-SPB stirrer ($2 \pm 2$ ng (h $10^6$ cells)$^{-1}$). The insulin profiles of the dynamic cultured INS-1 as spheroids using the 30°-3-SPB and the Rushton stirrer were similar to the insulin secretion of the INS-1 as a monolayer. INS-1, challenged as a monolayer, secreted 47 (basal) and 95 (acute) ng (h $10^6$ cells)$^{-1}$, which corresponded to SI = 2.1. In contrast, static produced spheroids (in cell-repellent 96-well plates) secreted 0.5 ng (h $10^6$ cells)$^{-1}$ within the acute phase and thereby secreted much lower insulin amounts.

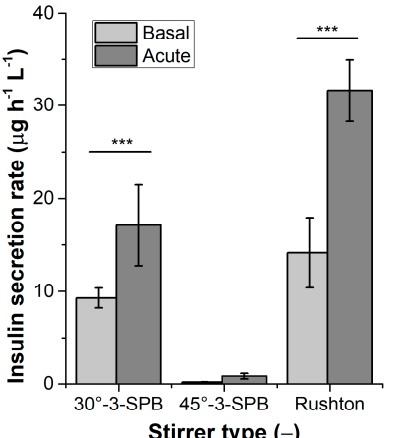 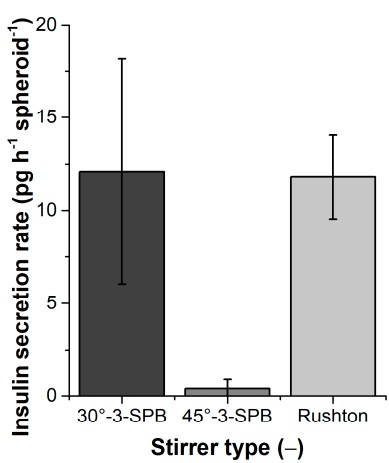 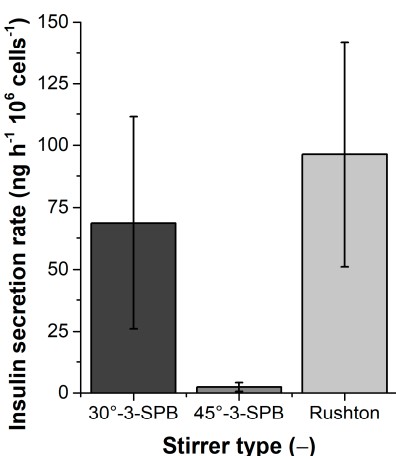

**Figure 7.** The functionality of β-cell spheroids determined using a glucose tolerance test. Left panel: the significant (*** <0.001, Student's *t*-test) increased secretion of insulin during the acute phase, compared with basal secretion, showed that β-cells produced with a 30°-3-SPB stirrer and a Rushton turbine responded to varying glucose concentrations. The insulin profiles of the β-cell spheroids generated with a 45°-3-SPB are only presented for completeness, but were not considered for discussion. Middle panel: the insulin secretion rate per spheroid. Right panel: we used a conversion factor to approximate the number of β-cells contained in a spheroid, to compensate for spheroid size differences. For each stirrer type (*n* = 2), three technical replicates of the glucose tolerance test were performed (error bars = STDV).

The reduced functionality of the static cultured spheroids was attributed to a limited mass transport within the spheroids and, consequently, low viabilities. Thereby, the STR-based production of β-cell spheroids restored the insulin secretion of the INS-1 from 2D cultures, while providing highly viable spheroids. The insulin profiles in static cultures only serve as a trend, which also applies to the insulin profiles of INS-1 cells in the literature, where static INS-1 spheroids showed a basal secretion of 20 $\mu$g L$^{-1}$, and an acute secretion of 40 $\mu$g L$^{-1}$, SI = 2 [10]. Static and dynamic spheroid formation vary widely, which may result in the differential compaction of the spheroids (cells per volume), and static spheroids often show necrotic cores, whereas those from dynamic cultures are often completely viable due to a better mass transfer and, thus, a more efficient transport of insulin and glucose. The handling of spheroids also differs: whereas the glucose-stimulated insulin secretion assay can be applied to static cultures with spheroids alone and maybe only to one spheroid per well, the preparation of samples from dynamic cultures incorporates multiple spheroids, varying in size, along with single cells. This indicates the complexity of testing, and highlights the need for further investigation to determine the functionality of β-cells cultured under dynamic conditions.

Our STR-based process produced functional INS-1-cell spheroids with yields of $18 \times 10^6 \pm 2 \times 10^6$ spheroids L$^{-1}$ for the 30°-3-SPB stirrer, $14 \times 10^6 \pm 4 \times 10^6$ spheroids L$^{-1}$ for the 45°-3-SPB stirrer, and $22 \times 10^6 \pm 0.6 \times 10^6$ spheroids L$^{-1}$ for the Rushton turbine (Table 3). By converting the spheroid mass to islet equivalents (IEQ), based on a standard islet of Langerhans' diameter of 150 $\mu$m and standard procedures for the transplantation of whole pancreatic islets [20], we calculated the IEQ counts for the complete STR volume (1 L) of $1.4 \times 10^6 \pm 0.6 \times 10^6$ for the 30°-3-SPB stirrer, $0.3 \times 10^6 \pm 0.2 \times 10^6$ for the 45°-3-SPB stirrer, and $1.1 \times 10^6 \pm 0.4 \times 10^6$ for the Rushton turbine (Table 3). Accordingly, one STR can produce sufficient spheroids for β-cell replacement therapies, given that the Edmonton protocol for islet transplantation requires $0.5–1 \times 10^6$ IEQs from up to three donors to achieve a post-transplantation insulin independence [22,63], and exceeds the yield of other culture systems (compare Table 1). Based on the high spheroid formation efficiency, the narrow spheroid distribution in combination with low single-cell counts, high viability and insulin secretion, and sufficient spheroid/IEQ yield, we recommend the 30°-3-SPB for the production of β-cell spheroids on a larger scale. In an earlier study, more than 1000 spheroids in the size range of 100–250 $\mu$m were produced from the β-cell line MIN6 [22]. The authors used a clinostat to simulate microgravity, and induced the self-assembly of spheroids, whose size was semi-regulated by adapting the seeding density (higher cell concentrations produced smaller spheroids). The MIN6-derived spheroids secreted more insulin than monolayers, and expressed functionally relevant genes, such as *insulin-2*, *glucokinase*, *SETD1A*, and *Kir6.2* at higher levels. The spheroids were also much more therapeutically effective than single cells, following a transplantation in a streptozotocin-induced diabetic mouse model. It is likely that spheroids perform better than single cells, not only because they are functionally superior, but also because single cells injected into the portal vein are too small for retention in the liver and most cells are lost, with their ultimate fate unclear. In contrast, spheroids larger than ~40 $\mu$m are more likely to be retained in the liver vessels. Islets < 100 $\mu$m in diameter performed better [21] than large spheroids (>250 $\mu$m), which showed lower functionality [22], probably reflecting the limited diffusive supply of nutrients, leading to the formation of necrotic cells. This shows the importance of the manufacturing of β-cell spheroids within a defined size range, which can be achieved using our STR setup.

## 4. Conclusions

We successfully used shaking flasks as a screening platform to determine the key process parameters (e.g., seeding density, spheroid strength) for the spheroid formation, using the β-cell line INS-1 as model. We then transferred the dynamic spheroid formation to a fully-controlled and monitored STR. We determined the mean energy dissipation as a transfer criterion to regulate the size of the spheroids formed under dynamic conditions,

and investigated three different stirrer types to evaluate the effect of the stirrer design (and the associated forces). Using the same mean energy dissipation, the 30°-3-SPB stirrer was slightly better than the Rushton turbine in terms of spheroid size distribution, also reducing the dead single-cell count and increasing spheroid formation efficiency and biological performance (higher $\mu_{Vol}$ and viability), while maintaining similar insulin secretion properties. In contrast, the 45°-3-SPB stirrer achieved poor results in all these aspects. We developed an image-based protocol to determine spheroid size and viability, and implemented the inline monitoring of biomass. The resulting spheroids achieved a similar glucose-dependent insulin secretion as standard 2D cultures of INS-1 cells. The large-scale cultivation of INS-1-cells achieved spheroid counts of up to $22 \times 10^6 \pm 0.6 \times 10^6$, and the corresponding IEQ counts were sufficient for β-cell replacement therapy. Although primary β-cells do not proliferate in culture, the development of human β-cell lines and β-cell-like cells derived from iPSCs is progressing, and robust large-scale production processes are needed for both. Other cell types, such as human mesenchymal stem cells (MSCs), can also benefit from shear-guided spheroid formation in an STR-based manufacturing process. Our STR-based spheroid formation process offers the scalability, process monitoring, and full control required for manufacturing, and allows the regulation of spheroid size while maintaining β-cell functionality.

**Supplementary Materials:** The following supporting information can be downloaded at: https://www.mdpi.com/article/10.3390/pr10050861/s1, Figure S1: Schematic description of (A): the mechanical power input of the stirrer, the resulting eddy cascade and energy dissipation, followed by the two spheroid stress concepts involving (B) tensile forces, (C) surface erosion, and finally (D) spheroid agglomeration, until the hydrodynamic forces are in balance with the adhesion force Fad; Table S1: Summary of stirrer properties and the resulting bioreactor geometry including the ratios for swept volume VS and working volume $V_L$, stirrer height $h_S$ and stirrer diameter $d_S$ or tank diameter $D_T$, as well as the bottom clearance $C_B = h_{S,bottom}/d_S$ ($h_{s,bottom}$ = installation height of stirrer from the bottom). References [11,41–46,53,55,56,64–67] are cited in the supplementary materials.

**Author Contributions:** F.P. conceived, designed and performed the experiments and wrote the paper. D.S. helped to draft and revise the manuscript, and supervised the research. All authors have read and agreed to the published version of the manuscript.

**Funding:** The work was funded by the Forschungscampus Mittelhessen (FCHM).

**Institutional Review Board Statement:** Not applicable.

**Informed Consent Statement:** Not applicable.

**Data Availability Statement:** All relevant data are given within the manuscript. The raw data presented in this study are available on request from the corresponding author.

**Acknowledgments:** We would like to thank Peter Czermak for the opportunity to work at his institute, to use his laboratories, additional funding, all the fruitful scientific discussions, and his input into this work. The authors would like to thank Richard M. Twyman and Catharine Meckel-Oschmann for language editing.

**Conflicts of Interest:** The authors declare no conflict of interest.

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
