# Peer review of "Large-Scale Production of Size-Adjusted β-Cell Spheroids in a Fully Controlled Stirred-Tank Reactor"

_processes, doi:10.3390/pr10050861_

Round 1

Reviewer 1 Report

Dear authors,

I enjoyed reading your manuscript.

It is very interesting, clearly written and of great importance for proper manufacture of β-cell spheroids and later for bringing the therapeutics into clinical practice.

The introduction provides enough insight into the subject.

The methods and results are well written and in detail.

The tables and figures clearly presented the data.

The literature used is appropriate and conclusion is consistent with the results obtained in the extensive research.

I am looking forward to reading about your next research.

Reviewer 2 Report

The authors of study studied the possibility of addressing β-cell spheroid manufacturing by a stirred-tank bioreactor process. They used INS-1 β-cell line as a model for process development. The study needs Major revisions prior to the publication.

Some of the sentences are unclear. The authors should improve the language of the manuscript.

The introduction section is really long for a research article. I suggest to shorten the introduction to one page. Some information might be provided in supplementary file.

Table 2 can be given as supplementary.

Information related to statistical analysis should be given under relevant figures.

The discussion section should be substantially improved by adding some more up to date studies and comparing your results with literature.

Round 2

Reviewer 2 Report

The article can be published in its current form.